# Assessing the Prospect of Joint Exploitations of Offshore Wind, Wave, and Tidal Stream Energy in the Adjacent Waters of China

**Zhan Lian [1,2,*], Weiye Yu [1] and Jianting Du [3]**

1   Institute of Marine Science, Guangdong Provincial Key Laboratory of Marine Disaster Prediction
    and Prevention, Shantou University, Shantou 515063, China
2   State Key Laboratory of Marine Environmental Science, Xiamen University, Xiamen 361102, China
3   First Institute of Oceanography, Ministry of Natural Resources, Qingdao 266061, China
*   Correspondence: zhanlian@stu.edu.cn

**Abstract:** The exploitation of marine renewable energy sources, such as offshore wind (OW), wave (WA), and tidal stream (TS) energy, is essential to reducing carbon emissions in China. Here, we demonstrate that a well-designed deployment of OW-WA-TS joint exploitation would be better than OW alone in improving performance in terms of the total amount and temporal stability of integrated power output in the northern Bohai Sea/Strait, the Subei Shoal, and the surrounding areas of Taiwan and Hainan Island. The design principles for an efficient joint energy deployment can be summarized as follows: first, a small ratio of WA output favors a temporally stable performance, except for areas around Taiwan Island and southwest of Hainan Island. Second, more TS turbines will contribute to steadier integrated outputs. Meanwhile, in the coastal waters of Guangdong and Zhejiang, the potential of WA to increase the total amount of power output is very high due to its minor impact on temporal stability. Finally, joint exploitation significantly reduces diurnal power fluctuations compared with OW alone, which is crucial for the steady operation of power grids, power sufficiency, and controllability in periods with low or no wind.

**Keywords:** offshore wind energy; wave energy; tidal stream energy; marine renewable energy; joint exploitation

## 1. Introduction

The exploitation of marine renewable energy (MRE) sources, such as offshore wind (OW), wave (WA), and tidal stream (TS) energy, is essential to mitigate the climate crisis [1,2]. Under the Paris Agreement, global society is turning toward MRE exploitations. OW, WA, and TS energy yields are all at different commercial stages. The installed capacity of OW has expanded exponentially over the last decades; it passed 56 GW in 2021, representing 7% of the total global cumulative wind capacity [3]. So far, the technologies of WA and TS energy converters are less mature than those for OW turbines [4,5]. However, there are many pilot projects to test prototypes of the generators and experimentally harvest energy [6,7]. Owing to the accumulation of technologies, we are now on the eve of soaring WA and TS power generation [8–10].

Despite the massive potential of OW energy, its practical use is limited by temporal variability [11,12]. The associated intermittency of OW power output hinders the electricity grid's stability [13]. WA and TS energy are not synchronized with OW, and each energy source might be complemented by the others; thus, joint exploitation could offer steadier power output. Additionally, the co-location of the various power generators can reduce costs and improve efficiency by sharing grid connections and infrastructure [14,15]. WA and TS power can be transported by these connections in periods with low OW output, which means an increase in infrastructure utilization.

Studies of OW, WA, and TS energy in the waters adjacent to China (Figure 1) have already proved fruitful. The annual mean power density of OW and WA increases from

north to south and from nearshore to offshore along China's coastline [16,17]. High power density is mainly found in the Taiwan Strait [18,19]. Both of these resources have obvious seasonality [20] and the TS resource is unevenly distributed [21]. The potential resources of key channels are more than half of the total TS energy in China's nearshore waters [4,22]. Table 1 lists the total potential MRE resources classified by province and marginal sea. OW and TS power were calculated by multiplying the surface power density (W/m$^2$) and the area of regions available for energy harvesting, while WA power was obtained by multiplying the line power density (W/m) and the coastline length.

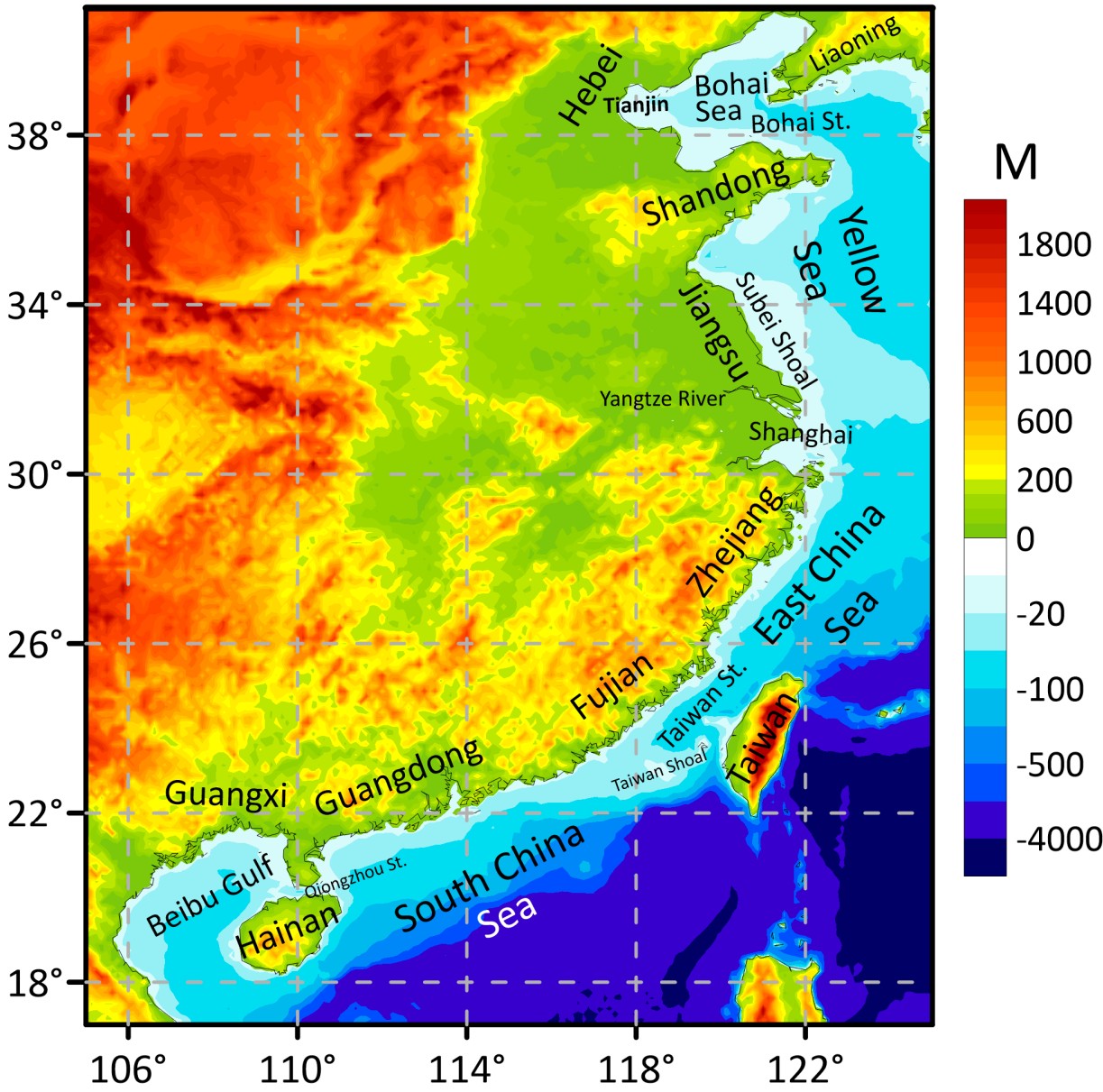

**Figure 1.** Topography of the study region (the data is from ETOPO global relief model, https://www.ncei.noaa.gov/products/etopo-global-relief-model accessed on 27 January 2023).

**Table 1.** Statistics of exploitable OW [23], WA [16], and TS [21] resources classified by province and marginal sea. (TS power only includes limited nearshore channels due to a lack of observations.)

| Province | OW Power GW | WA Power MW | Marginal Sea | TS Power GW |
|---|---|---|---|---|
| Liaoning | 60.6 | 184.6 | Bohai and Yellow Sea | ~1.1 |
| Hebei | 24.1 | 99.5 | | |
| Tianjin | 5.6 | 13.7 | | |
| Shandong | 76.5 | 483.8 | | |
| Jiangsu | 107.6 | 94.3 | | |
| Shanghai | 24.3 | 160.1 | East China Sea | ~4.6 |
| Zhejiang | 53.8 | 1916.0 | | |
| Fujian | 28.1 | 2910.7 | | |
| Guangdong | 51.7 | 4557.2 | South China Sea | ~0.4 |
| Guangxi | 26.6 | 81.1 | | |
| Hainan | 10.4 | 4204.9 | | |

The development of OW, WA, and TS commercial exploitations is varied in China. China has become one of the fastest-growing countries in OW power development [24], installing 80% of all new global OW energy capacity in 2021 [3]. Although the WA and TS industries lag relatively behind OW, there has been steady development. For instance, many full-scale tests of WA energy harvesting have been performed in China [10]. A TS power generator with a rated power of 3.4 MW has been successfully connected to the grid in Zhejiang Province, marking the start of commercial operations of TS exploitation in China [6].

The joint exploitation of OW, WA, and TS energy will play an important role in the future MRE harvesting strategy of China for the following reasons. First, it is necessary for China to reach its carbon-neutral target. China has promised to begin cutting its carbon emissions within the next ten years and to become carbon neutral before 2060. The growth potential of OW exploitations to achieve this goal is tremendous. Hence, the comprehensive and efficient utilization of OW farms is a high priority. Second, joint exploitations are expected to assist in leveling the diurnal variabilities of OW, which are significant in the areas studied [13]. This is because diurnal WA fluctuations lag behind OW due to larger inertia, while the TS mainly varies diurnally. These diurnal variations have a profound impact on the stability of the power grid [25]. They have a larger magnitude and less predictability than low-frequency (such as seasonal) fluctuations, and hence are the focus of OW power forecasting systems [26].

So far, few studies have investigated triple-energy joint harvesting in waters adjacent to China, and the complementarity of multiple energy resources within the diurnal band has not been fully revealed. Some studies have assessed the potential of OW and WA [27–29], OW and TS [30], and OW and solar [31] in the USA and Europe. The first assessment of their joint potential in China's nearshore waters was performed in the South China Sea [32]. Although that study did not consider in detail the impact of the OW-WA hybrid on the total output, the results revealed three promising areas for hybrid power generation in the Taiwan Strait, Luzon Strait, and to the southeast of the Indochina Peninsula. Recently, the complementarity and synergy of OW and WA energy in the southern China coastal regions has been investigated [33]. It was revealed that the correlation between the two resources varies geographically, and the best sites for future joint exploitation are along the east China coast. It should be noted that the study did not include the northern waters adjacent to China, and the TS effect was neglected.

In summary, to develop an efficient multiple energy exploitation plan that will attract investors, several questions urgently need to be answered about the waters adjacent to

China. Where is joint exploitation particularly promising? How could the exploitation be made highly efficient? To what extent would WA and TS energy flatten the diurnal variabilities of OW? How steady would the integrated power outputs eventually be? This study aims to address these concerns. The three resources were diagnosed by using reanalysis data and numerical simulations. The data and methods are described in Section 2. In Section 3, we analyze the results and describe the relationship among the three energy sources in the adjacent waters of China. Finally, the conclusions are presented in Section 4.

## 2. Data and Methods

We used ECMWF Reanalysis v5 (ERA5) to calculate the OW and WA energy. The ERA5 dataset provides global atmospheric and oceanic data (hourly, ~31 km spatial resolutions) using 4D-Var data assimilation (for more details, see https://www.ecmwf.int/en/forecasts/dataset/ecmwf-reanalysis-v5, accessed on 1 July 2022). The quality of the data has been proven and it has been widely applied in the study of MRE [34–37]. The tidal current data was obtained from TPXO 9 [38]. It is a barotropic tide model and presents tidal elevations and horizontal transport (~18 km spatial resolutions). Satellite altimetry and in situ observations are assimilated to improve the simulation accuracy (for more details, see https://www.tpxo.net/global/tpxo9-atlas, accessed on 15 September 2020). Due to the different spatial resolutions of the two datasets, we interpolated all elements into a uniform grid (1/8 × 1/8 degrees). We used data from 2021 to conduct the following analyses.

The OW power output from wind turbines ($OW_O$) was determined by the cut-in/out of wind speed, the rated power, and the performance of a specific wind turbine. When the wind speed was less or greater than the cut-in or cut-out speed, the $OW_O$ was zero or capped by the rated power. A wind turbine (SWT-6.0-154) developed by Siemens Gamesa (SGRE) was selected for this study. This kind of device has been widely installed in the China Sea, and the parameters are similar to those of other mature devices [33]. When the wind speed was between the cut-in and cut-out speeds, the $OW_O$ was obtained from the power curve of the SWT-6.0-154 (Figure 2).

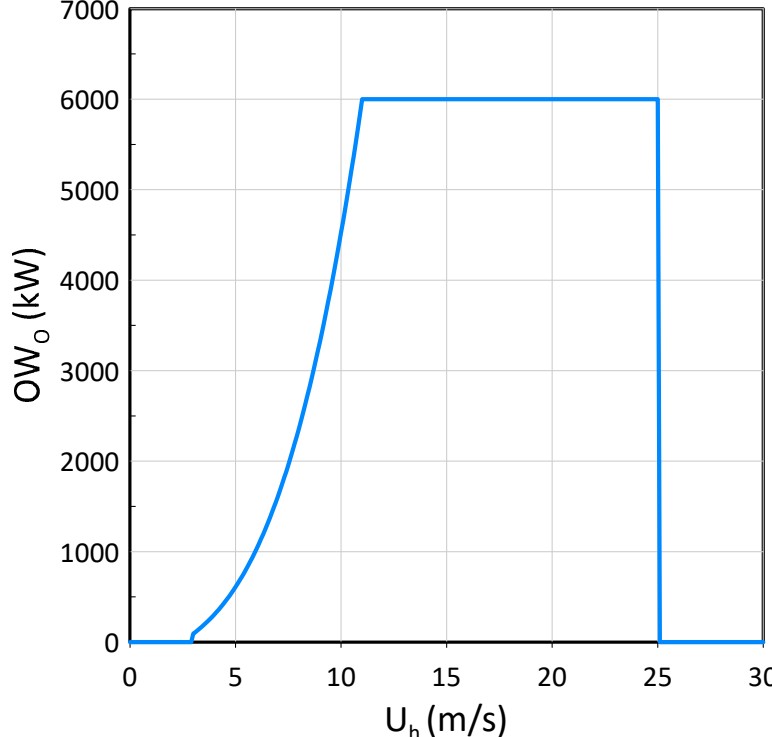

**Figure 2.** Power curve of SWT-6.0-154 [33].

The curve shows that the $OW_O$ varied according to the wind speed at hub height ($U_h$). We extrapolated the ERA5 wind field ($U_{ERA}$, at 10 m) to the hub height (100 m) by the power law [39]. The extrapolation is

$$U_h = U_{ERA} \left( \frac{Z_2}{Z_1} \right)^a$$

where $Z_1$ is 10 m and $Z_2$ is 100 m. $a$ is the wind shear exponent, and $a = 0.143$ in this study [40].

The WA power output ($WA_O$) from a generator was determined by the performance of the device, the significant wave height ($H_S$), and energy period ($T_E$). We selected a floating, slack-moored wave energy converter named Wave Dragon to study the spatial–temporal variations of $WA_O$. This wave energy converter has a rated power (5900 kW) very close to that of the SWT-6.0-154, and it has been recommended for wave energy extraction in offshore areas [41]. The power matrix of the Wave Dragon is shown in Figure 3.

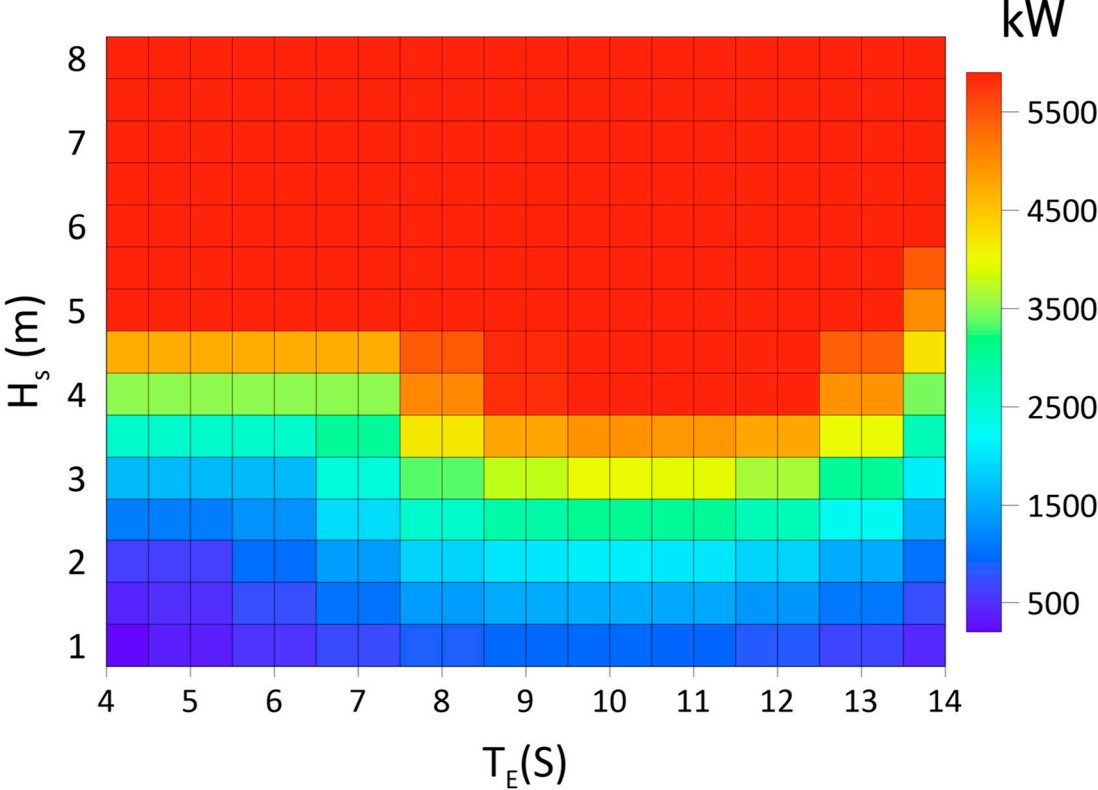

**Figure 3.** Power matrix of Wave Dragon [42].

There are other types of WA converters [43], but the output power of shallow-water-type WA converters is much less than that of the Wave Dragon, and their output is highly correlated with that of the Wave Dragon, especially in nearshore waters. Most OW farms are located in water far away from shorelines; therefore, the shoreline-type WA converter is not suitable for being hybrid-deployed with OW farms. Consequently, this study was based on the output power of the Wave Dragon. The characteristics of the outputs from shallow-water-type WA converters are presented in Appendix A (Figures A1 and A2).

The output power of a TS turbine ($TS_O$) can be described mathematically as follows:

$$TS_O = \frac{1}{2} \rho A C_P U^3$$

where $\rho$ is the seawater density and $U$ is the tidal current speed. $A$ is the swept area of the TS turbine, and the radius of the turbine is 10 m in this study. $C_P$ is the energy conversion rate of turbines. According to the performances of various commercial TS turbines [44,45], we set $C_P = 0.4$ in this study. The output is zero if the current speed is less than 0.5 m/s.

We designed a series of scenarios to investigate the geographical and temporal variations of integrated output ($P$) from multiple-source joint MRE exploitation in China's adjacent waters (Table 2). In each scenario, $P$ was calculated as follows:

$$P = OW_O + C_1 \cdot WA_O + C_2 \cdot TS_O$$

$C_1$ and $C_2$ represent the numbers of WA converters and TS turbines, respectively, which are jointly installed with an OW turbine. The values of these two factors for all scenarios are listed in Table 1. The maximum $C_1$ and $C_2$ were 4 and 10, respectively. This extreme installation scheme is not necessarily impractical given the fact that the required space for a WA converter [46] or for a TS turbine [15] is much smaller than that for an OW turbine.

**Table 2.** Configurations of $C_1$ and $C_2$ in all scenarios. (If $C_1$ is less than 1, it means one WA converter for multiple OW turbines. For example, $C_1 = 0.25$ means 1 WA converter for 4 OW turbines.)

|  | $C_1$ | $C_2$ |  | $C_1$ | $C_2$ |
|---|---|---|---|---|---|
| Scenario 1 | 0.25 | 1 | Scenario 16 | 1 | 6 |
| Scenario 2 | 0.25 | 2 | Scenario 17 | 1 | 8 |
| Scenario 3 | 0.25 | 4 | Scenario 18 | 1 | 10 |
| Scenario 4 | 0.25 | 6 | Scenario 19 | 2 | 1 |
| Scenario 5 | 0.25 | 8 | Scenario 20 | 2 | 2 |
| Scenario 6 | 0.25 | 10 | Scenario 21 | 2 | 4 |
| Scenario 7 | 0.5 | 1 | Scenario 22 | 2 | 6 |
| Scenario 8 | 0.5 | 2 | Scenario 23 | 2 | 8 |
| Scenario 9 | 0.5 | 4 | Scenario 24 | 2 | 10 |
| Scenario 10 | 0.5 | 6 | Scenario 25 | 4 | 1 |
| Scenario 11 | 0.5 | 8 | Scenario 26 | 4 | 2 |
| Scenario 12 | 0.5 | 10 | Scenario 27 | 4 | 4 |
| Scenario 13 | 1 | 1 | Scenario 28 | 4 | 6 |
| Scenario 14 | 1 | 2 | Scenario 29 | 4 | 8 |
| Scenario 15 | 1 | 4 | Scenario 30 | 4 | 10 |

We used the coefficient of variation ($V$) to evaluate the temporal changes of $P$. $V$ is expressed as

$$V = \frac{\sigma}{\mu}$$

where $\sigma$ is the standard deviation of the time series of $P$, and $\mu$ is the time mean.

## 3. Results

First, we investigated the outputs from individual energy exploitations. We then analyzed the most energy-efficient scenarios for joint exploitations. These scenarios were defined as the ones reaching the highest $\mu$ at the lowest $V$.

### 3.1. Output from Single Energy Exploitations

First, we separately calculated the time-averaged outputs from OW turbines, WA converters, and TS turbines. In order to make the values of the three outputs comparable, $C_1$ and $C_2$ were set at 4 and 10, respectively. The $OW_O$ was the largest component in the

nearshore waters (Figure 4). Its geographical variation was not as remarkable as the other energy sources. The maximum $\mu$ of $OW_O$ occurred in the Taiwan Strait and the Luzon Strait due to topographic effects. The $WA_O$ was larger in offshore areas than the nearshore areas. The $\mu$ of $C_1 \cdot WA_O$ was less than 2 MW in most areas to the north of the Yangtze River, in the Taiwan Strait, and in the Beibu Gulf. In other areas, the high $C_1 \cdot WA_O$ (>2 MW) regions were close to the coastline. Its value reached more than 4 MW, which was larger than the $OW_O$ in those areas, to the east of Taiwan Island and in the central basin of the South China Sea. Contrary to the other two types of MRE, the $\mu$ of $C_2 \cdot TS_O$ was less than 0.3 MW in most of the studied waters. The $TS_O$ was zero in the center of the Yellow Sea, to the east of Taiwan Island, and in the continental shelf area of the northern South China Sea, since the maximum tidal current speed is less than 0.5 m/s in those places. There were several high-$TS_O$ hotspots, which were the waters in the northern Bohai Sea/Strait, in the Subei Shoal, in the Taiwan Strait, and in the Qiongzhou Strait. The maximum $\mu$ of $C_2 \cdot TS_O$ was about 1 MW at these hotspots.

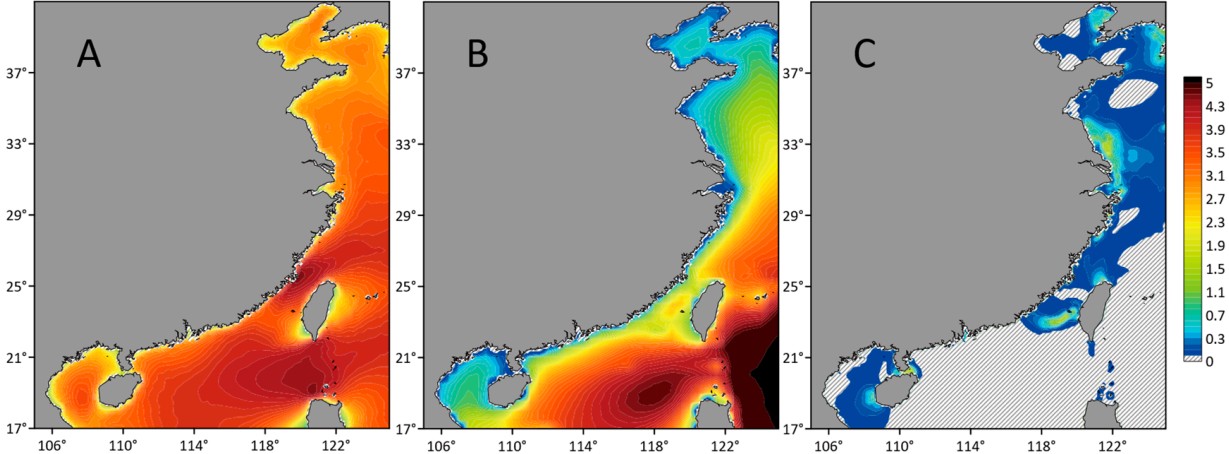

**Figure 4.** Distribution of the $\mu$ (MW) of $OW_O$ (**A**), $WA_O$ (**B**), and $TS_O$ (**C**).

The time stabilities of the three energy components were obviously different (Figure 5). The spatial averaged $V$ of $OW_O$ was the lowest, and the lower $V$ appeared in the deeper regions. The $V$ of $WA_O$ was significantly higher than for $OW_O$ in most waters. The $WA_O$ was more temporally stable in the nearshore waters which are close to open water, such as along the Zhejiang and Guangdong coasts. The $V$ of $TS_O$ was around 1 at the $TS_O$ hotspots, which was comparable with that of $OW_O$, but less than for $WA_O$.

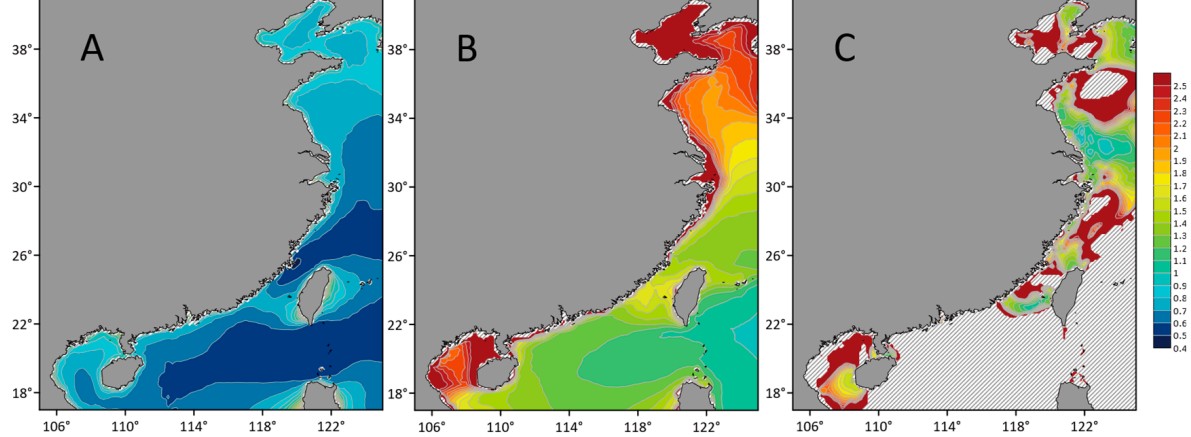

**Figure 5.** Distribution of the $V$ (dimensionless) of $OW_O$ (**A**), $WA_O$ (**B**), and $TS_O$ (**C**).

The temporal correlation between various energy types is important for the performance of joint energy harvesting. A lower correlation coefficient ($R$) implied that the two types of energy had high complementarity. The $R$ from the three energy pairs showed remarkable spatial variations (Figure 6). The $WA_O$ was relatively less correlated with $OW_O$ in the Bohai Sea, the East China Sea, the Beibu Gulf, and the northern Taiwan Strait. The averaged $R$ was about 0.4 in those regions. The $TS_O$ was generally uncorrelated with $WA_O$ and $OW_O$ ($|R| < 0.05$).

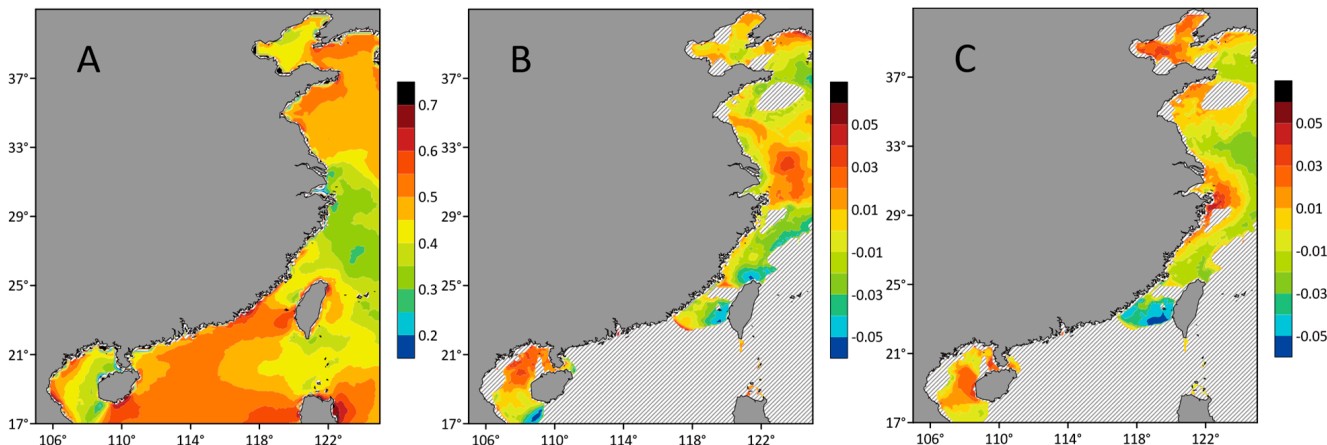

**Figure 6.** Distribution of the temporal correlation coefficient between $OW_O$ and $WA_O$ (**A**), $TS_O$ and $OW_O$ (**B**), $TS_O$ and $WA_O$ (**C**). Note the color bars are different.

### 3.2. Output from Joint Exploitations for the Least Possible V

We examined the outputs from double-energy and triple-energy exploitations. The double-energy exploitations referred to OW energy extraction combined either with WA or TS. The scenario indexes were defined as Scenarios 1 to 6, where there were 0.25, 0.5, 1, 2, or 4 WA converters, or 1, 2, 4, 6, 8, or 10 TS turbines, jointly deployed with 1 OW turbine. The indexes for the triple-energy exploitation scenarios are shown in Table 1.

We selected the minimum $V$ of the $P$ from all the scenarios and recorded the indexes of the scenarios in which the minimum $V$ was reached (Figure 7), and the relative changes of $\mu$ (Figure 8a–c) and $V$ (Figure 8d–f) compared with $OW_O$. For OW-WA joint exploitation, Scenario 1 reached the minimum $V$ in most study waters (Principle 1). This was because the WA and OW were relatively highly temporally correlated in these regions, and the $V$ of $WA_O$ was large. A low ratio for $WA_O$ in this joint exploitation, therefore, favored a temporally stable synergistic output. However, exceptions occurred to the east of Taiwan Island due to a large $WA_O$ there, and to the southwest of Taiwan Island and Hainan Island as a result of a small $OW_O$. From the Pacific to Taiwan Island, one OW turbine combined with 0.5, 1, or 2 WA converters facilitated a stable $P$. In terms of the OW-TS joint exploitation, the scenario with the most stable $P$ was Scenario 6 in all regions where the current speed was larger than the cutoff speed of the turbines, which meant that the stability of $P$ benefited the most from the combination of one OW turbine and 10 TS turbines (Principle 2). The combination was complementary due to the independence of OW and TS. The indexes for the minimum-$V$ triple-energy exploitations were mixtures of two double-energy exploitations. Principle 1 plus Principle 2 was the best strategy for an efficient $P$ (high $\mu$ and low $V$) from triple-energy exploitation. The descriptions of the best strategies are summarized in Table 3.

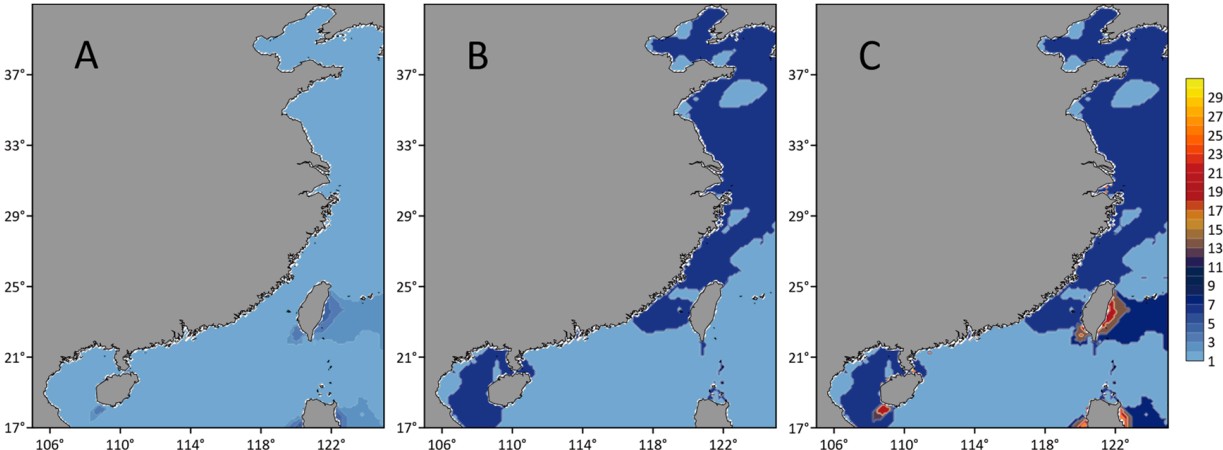

**Figure 7.** Distribution of the scenario indexes for OW-WA (**A**), OW-TS (**B**), and OW-WA-TS (**C**) exploitations. In these scenarios, the joint exploitation outputs hold the least *V*.

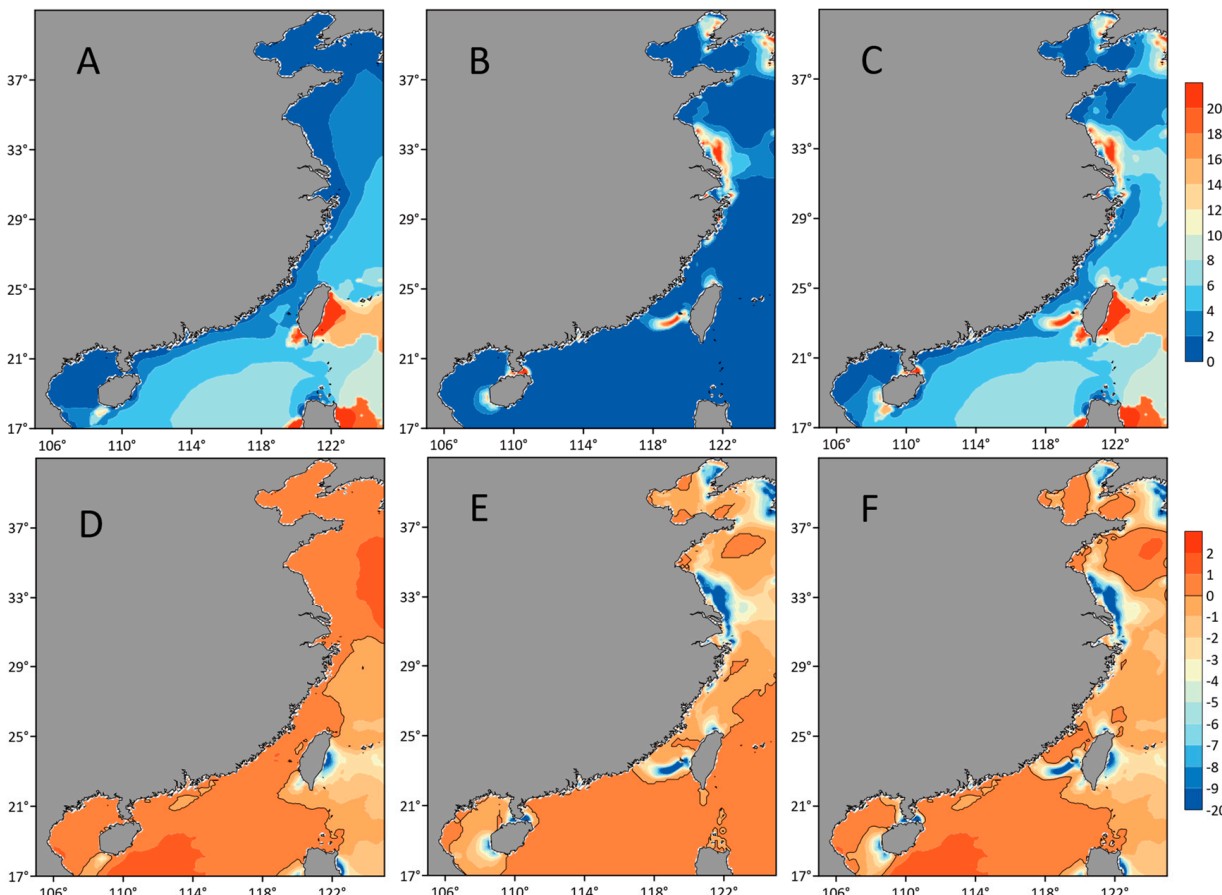

**Figure 8.** Distribution of the ratio (%) of the joint exploitation performances compared with $OW_O$. The joint exploitation of each geographical grid operates in the scenario indicated in Figure 6. $\mu$ (**A**) and *V* (**D**) of OW-WA, $\mu$ (**B**) and *V* (**E**) of OW-TS, and $\mu$ (**C**) and *V* (**F**) of OW-WA-TS exploitation.

**Table 3.** Descriptions of the best strategies for various joint exploitations.

|  | Description |
|---|---|
| OW-WA | 1 WA converter with 4 OW turbines in most regions |
| OW-TS | 1 OW turbine with 10 TS turbines |
| Triple-energy | Mixture of the above two strategies |

### 3.3. Output from Joint Exploitations for Enlarging μ

It should be noted that the relative increase of $\mu$ and decrease of $V$ were unbalanced in the "optimal" scheme for the minimum $V$. We defined the optimal scheme as the scheme which makes the integrated power output as temporally stable as possible. In that scheme, there are some regions, such as the coastal waters of Zhejiang and Guangdong, where the $V$ changed slightly (<1%), but the $\mu$ increased more obviously (from 2% to 10%). This suggested that the scheme pursuing the lowest $V$ suppresses the increased potential of the joint energy output in these seas.

We designed a new algorithm to find a scenario which is aggressive in enlarging $\mu$, while also tolerating a relatively moderate increase in $V$. The weighted factor ($V'$) was used in this algorithm. The index, $\mu$, and $V$ of the scenario in which $V'$ was at a minimum were recorded. $V'$ was expressed as

$$V' = \frac{\| \sigma \|}{10^{\|\mu\|}}$$

where $\| \|$ refers to the normalization of the variable to the maximum in the entire region and in all scenarios.

The differences between the scenarios (Figure 9) for the least $V'$ and $V$ were twofold. Firstly, along the coast of Guangdong and Zhejiang, the target number of WA converters changed from 0.25 to 0.5. In this condition, the enhancements of $\mu$ increased by ~6% to ~10% (Figure 10a), while the relative changes of $V$ were still ~2% (Figure 10b). Especially near the Peal River Estuary, the increase was larger than 14%. Secondly, to the southwest of Taiwan Island and Hainan Island, the area of the regions with high increased $\mu$ and the enhancements of $\mu$ were further enlarged; on the other hand, the $V$ in those regions did not obviously change compared with the scenario for the minimum $V$.

### 3.4. Joint Exploitation Performance in the Diurnal Band

Compared with longer-term fluctuations, such as seasonal variations, the high-frequency OW power variabilities such as the diurnal band have larger magnitudes and lower predictability. Moreover, the power demands have significant oscillations with a similar frequency. Therefore, it was necessary to analyze the joint exploitation performance throughout the day. The output power used in this study was by hourly interval; therefore, it was possible to assess the joint exploitation performance in the diurnal band.

To examine the diurnal variations of the three energy sources, we first used a high-pass filter to investigate the temporal stability of $OW_O$ and $WA_O$ in the high-frequency band. The variabilities with a period of less than 24 h were retained after filtering. We defined $V_{HP}$ as the standard deviation of the filtered output divided by the mean of the raw output. The results (Figure 11) revealed that $OW_O$ was temporally unsteady compared with $WA_O$ in the diurnal band. For the $OW_O$ in the Bohai Sea and the northern Yellow Sea, more than 50% of $V$ was caused by $V_{HP}$. The ratio decreased to ~30% in the nearshore waters to the south of the Yangtze River. To the southwest of Taiwan Island and Hainan Island, the ratio rose to 50% again. The associated geographical pattern of $WA_O$ was similar, but the value was much less than for $OW_O$. The maximum ratio was less than 45% in the northern waters, and it could reach as low as 20% near the Zhejiang, Fujian, and Guangdong coasts. This phenomenon was the result of the different inertness between atmospheric and marine oscillations. Due to the density discrepancy, the water oscillations hold a longer period and a lower frequency.

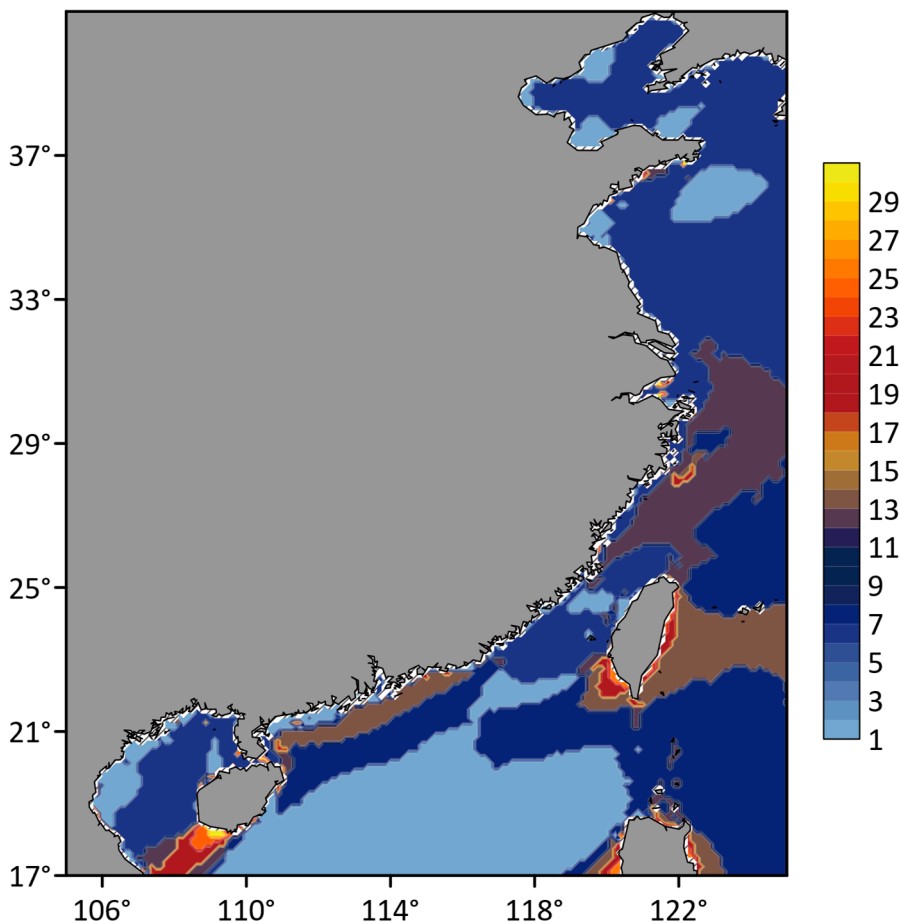

**Figure 9.** Distribution of the scenario indexes for OW-WA-TS exploitation in which the minimum $V'$ of $P$ occurs.

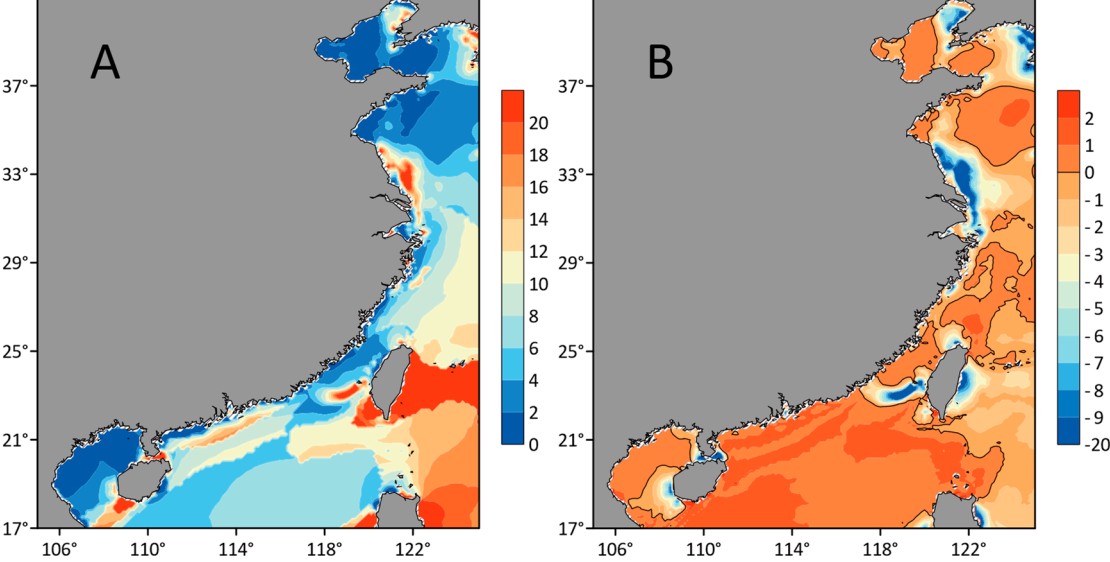

**Figure 10.** Distribution of the ratio (%) of the joint exploitation performances compared with $OW_O$. The exploitation uses the scenario index presented in Figure 8. $\mu$ (**A**) and $V$ (**B**) of the OW-WA-TS exploitation.



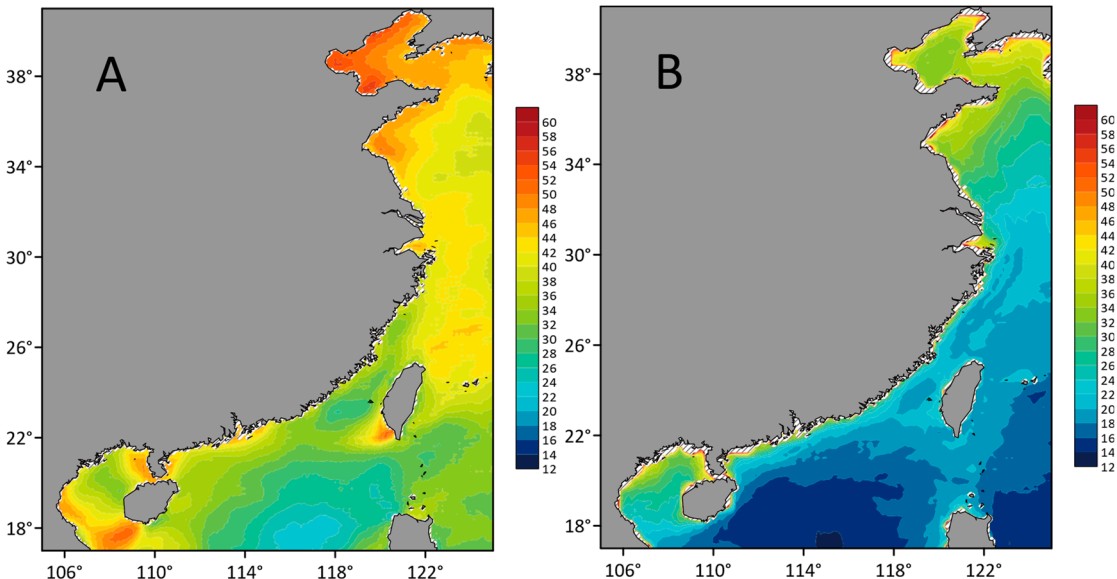

**Figure 11.** Distribution of the ratio (%) between the $V_{HP}$ and the raw $V$. $OW_O$ in (**A**), and $WA_O$ in (**B**).

We then calculated the changing rates of output power by averaging the absolute temporal changes at each geographical point, and the rates were normalized by the maximum output during the study period at this point. The result represented the averaged relative change compared to the maximum in one hour (Figure 12). The fastest rate was for $TS_O$, which can change from 0 to the rated power within 4~5 h in the Subei Shoal and the Taiwan Strait. This was consistent with the fact that semi-diurnal and diurnal tides dominate the study waters. To the north of the Yangtze River, $OW_O$ was able to change from zero to the rated power within one day (if the rate is faster than 4.2 %/h, it means that the output can change from 0 to 1 in 24 h). The slowest rate was found in the Taiwan Strait and the northern South China Sea. On the other hand, $WA_O$ changed much more slowly compared with $OW_O$, particularly along the southern and eastern coastlines.

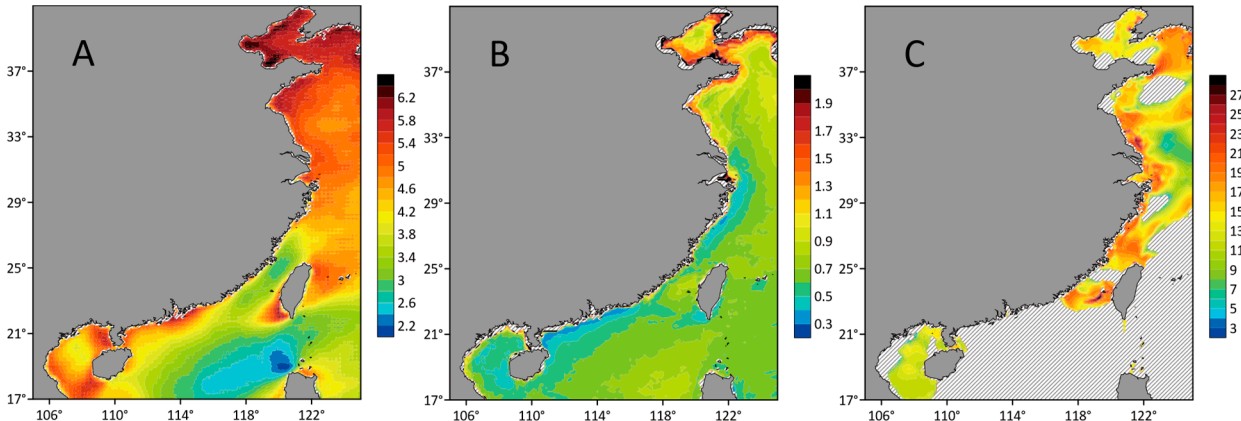

**Figure 12.** Distribution of the relative rates (%/h) of $OW_O$ (**A**), $WA_O$ (**B**), and $TS_O$ (**C**). Note the color bars are different.

According to the facts discussed above, joint exploitations could largely decrease the $V_{HP}$ of integrated outputs compared with separate OW energy harvesting (Figure 13). The improvements would be especially significant with OW-WA exploitations. In the nearshore waters of Jiangsu, Zhejiang, and Fujian, the least possible $V_{HP}$ from joint exploitations was less than that the $V_{HP}$ of $OW_O$ by ~20%. The ratio further increased to more than 30% near the Guangdong coast and to the south and east of Hainan Island. To the east and

southwest of Taiwan Island, the ratio could reach 50%. TS cannot decrease the $V_{HP}$ as much as WA. The most remarkable decrease induced by TS occurred in the northern Bohai, the Subei Shoal, the Taiwan Shoal, and to the southwest of Hainan Island. In these places, the maximum and average values of decrease were 6% and 3%, respectively. Due to the lower contributions from TS, the least possible $V_{HP}$ from OW-WA-TS joint exploitations had patterns similar to those of OW-WA exploitations.

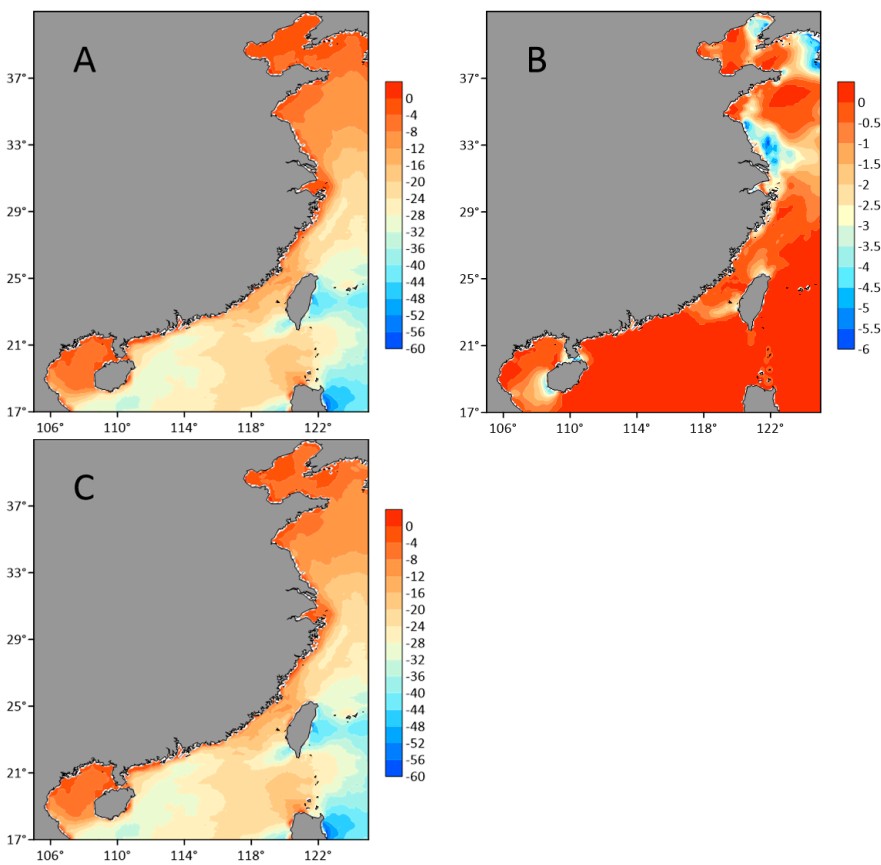

**Figure 13.** Distribution of the ratio (%) between the least possible $V_{HP}$ from joint exploitations and the $V_{HP}$ from $OW_O$. OW-WA in (**A**), OW-TS in (**B**), and OW-WA-TS in (**C**). Note the different color bars.

## 4. Conclusions

The investigations of this study revealed that it would be possible, and beneficial for the grid operation, to increase the total amount of integrated power output and simultaneously decrease its temporal variations by a joint exploitation of offshore wind (OW), wave (WA), and tidal stream (TS) energy in the adjacent waters of China. Furthermore, the comprehensive synergy and reduced fluctuation of power output from this triple-source energy exploitation would be better than with either double-source OW-WA or OW-TS generation. The most remarkable improvements in the power output would be in the northern Bohai Sea/Strait, the Subei Shoal, and the surrounding areas of Taiwan and Hainan Island. The total amount and temporal stability, represented by the time mean ($\mu$) and the variation coefficient ($V$), in these regions can increase and decrease, respectively, by about 20%. In the scenario framework of this study, to achieve the least $V$ of the integrated energy output, the ideal strategy for the MRE joint exploitation can be summarized as follows:

Principle 1: A small ratio of $WA_O$ favors a temporally stable $P$, except for areas around Taiwan Island, and to the southwest of Hainan Island.

Principle 2: More TS turbines contribute to steadier integrated power outputs.

For the coastal waters of Guangdong and Zhejiang, the potential of WA to increase the total amount of power output is large, due to its minor impact on temporal stability. In

these areas, it is possible to design a joint exploitation that largely enhances the total power outputs without significantly changing the stability.

In the study regions, more than 30% of the temporal variability of OW was attributed to diurnal oscillations. This ratio was much lower in the variation of WA. Therefore, WA-OW, but not TS-OW, exploitations can significantly smooth the diurnal variabilities of integrated power outputs. The discrepancy was obvious in the southeastern waters adjacent to China.

In this study, we assumed that the maximum number of TS turbines co-located with one OW turbine would be 10. More TS turbines may mathematically further improve the performance of the joint energy yield; however, it might be impractical and uneconomical in terms of marine engineering. Therefore, analysis of the effect of having more than 10 TS turbines was beyond the scope of this research. The upper threshold for the number of WA converters was selected based on a similar reason. Additionally, more WA converters would further decrease the stability of the total output of joint exploitation.

These ratios in the hybrid deployment would be workable due to the fact that the spatial requirements of OW turbines are much larger than for tidal turbines and wave converters, and there needs to be considerable vacant space between OW turbines in operating OW farms. Moreover, the scale of OW turbines and farms will increase in the future [47]. Therefore, it was reasonable to use 10 [15] and 4 [46] as the maximums for tidal turbines and wave converters, and these were sufficient to address the issues with which this study was concerned.

Related issues in the marine and electrical engineering fields, such as the grid connection of joint systems, the arrangement of various generators, as well as the construction, operation, and maintenance of joint MRE farms, were not included in this study.

This study presented an assessment of the potential of joint MRE harvesting in the adjacent waters of China. After pinpointing the regions with high potential, observations and simulations with fine scales will now be needed by conducting cross-disciplinary investigations involving both geophysical science and marine engineering. This study is a stepping stone to attract investors and plan MRE strategies in the study areas.

**Author Contributions:** Z.L.: conceptualization, methodology, writing—original draft preparation, formal analysis, visualization, project administration, and funding acquisition. W.Y.: data curation, software, and validation. J.D.: conceptualization, writing—reviewing and editing, and funding acquisition. All authors have read and agreed to the published version of the manuscript.

**Funding:** This work was supported by the Basic Scientific Fund for National Public Research Institutes of China (grant number 2019Q06); the Shantou University Scientific Research Funded Project (grant number NTF21009); the MEL Visiting Fellowship (grant number MELRS2118); the Natural Science Foundation of China grants (grant number 41906028); and the National Key R&D Program of China (grant number 2022YFC3004200).

**Institutional Review Board Statement:** Not applicable.

**Informed Consent Statement:** Not applicable.

**Data Availability Statement:** No new data were created or analyzed in this study. Data sharing is not applicable to this article.

**Conflicts of Interest:** The authors declare no conflict of interest.

## Appendix A

The characteristics of the outputs from shallow-water-type WA converters are presented in Figures A1 and A2.

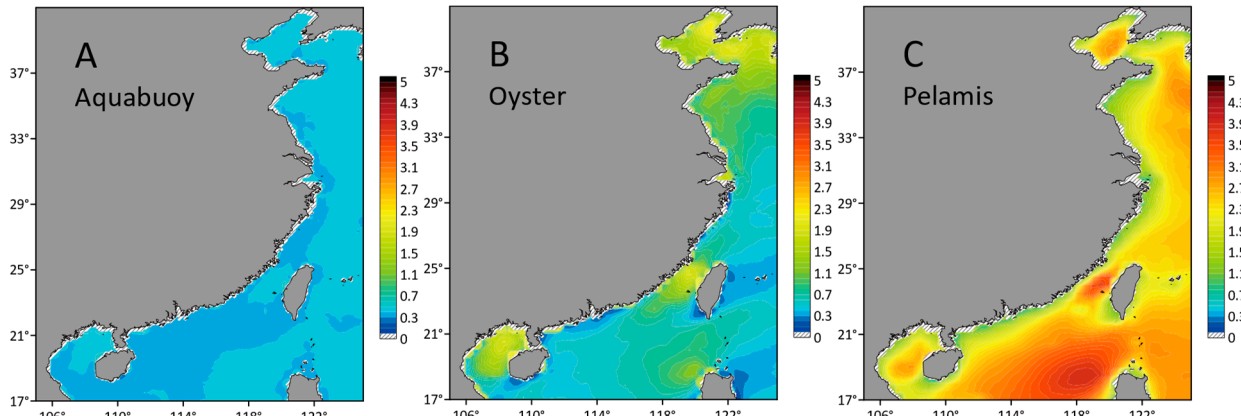

**Figure A1.** Distribution of the $\mu$ (MW) of $WA_O$ from various WA converters. The power matrix was obtained from reference [43]. The values in (**A**,**B**) and in (**C**) are enlarged by 5 and 10 times, respectively, to have the same scales as $WA_O$ from Wave Dragon.

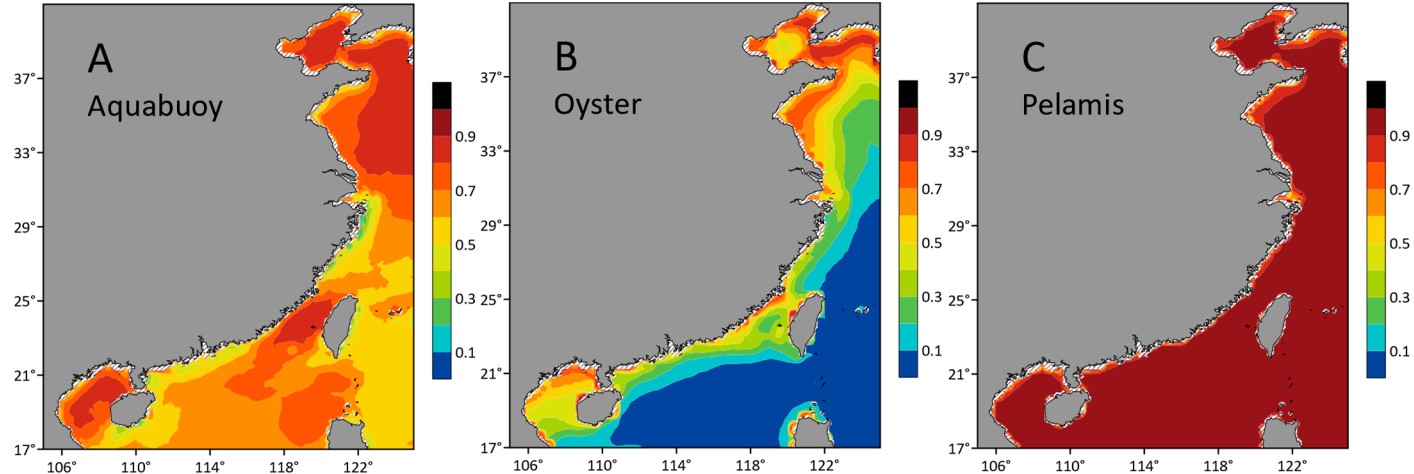

**Figure A2.** Distribution of the temporal correlation coefficient between various WA converters and Wave Dragon.

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
