# Peer review of "Assessing the Prospect of Joint Exploitations of Offshore Wind, Wave, and Tidal Stream Energy in the Adjacent Waters of China"

_jmse, doi:10.3390/jmse11030529_

Round 1

Reviewer 1 Report

The manuscript is well-written and relevant. 

The work has certain simplifications. For example, it does not consider the grid connection difficulties of these kinds of joint systems. It may be interesting for readers to know which points the authors considered and which ones they have left out.

There are a few minor language errors that may need corrections. For instance, page 5 above Table one WA convertor is probably WA converter. On page 6, below Figure 4, "components are obvious different" is probably "obviously different" and so on.

Reviewer 2 Report

-Missing reference for Figure1.

-Which year are referred the data from ERA5?

Reviewer 3 Report

In general, please consider using "energy yielding" instead of "mining". 

1) In Abstract:

In this sentense: "Here, for the first time, we demonstrate that a well-designed deployment can make the performance of the OW-WATS joint exploitation better than OW alone, in terms of the total amount and the temporal stability of the integrated power output in the northern Bohai Sea/Strait, the Subei Shoal, and the surrounding areas of Taiwan and Hainan Island. " - will you please reformulate that "for the first time" refers better to the mentioned geographical areas. Otherwise, it might give a wrong impression as "for the first time" in general. 

2) In Abstract:

In this sentense: " Finally, the joint exploitation significantly reduces diurnal power fluctuations compared with individual OW, which is very important for the steady operation of power grids." - you could add one more benefit which is better power sufficiency and contrallability in periods with low or no wind.

3) Page one: "The annual mean power density of OW and WA increases from the north to the south and from the nearshore to the offshore along China’s coastline[16,17]. The high power density is mainly found in the Taiwan Strait[18,19]. " - Here it would be very interesting to add a Table with some power density numbers. If you are able to include such a Table, please do it, it will benefit your paper. 

4) In Page 1: " Additionally, the co-locations of the various power generators reduce cost and improve efficiency by sharing grid connections and infrastructure[14,15]." - Here it is important to add that increased efficiency of the connections is due to power transport from WA and TS in periods with low OW output. Meaning that the connections are used all the time though the OW output can be low in periods. 

5) In Page 2: In this sentense " It is because high-frequency WA fluctuations" - Please provide typical periods of such fluctuations. The term "high-frequency" seems misleading in this context. 

6) In Page 2: Here "and OW and solar[30] in America" - Please use USA instead of America. 

7) In Page 3: "ECMWF Reanalysis v5 (ERA5) " - Please add if this software is commercially available or developed in-house. If commercially available, then please include the vendor. It is important to know if the described method can be applied by others. 

8) The same comment as above to this "The tidal current data was obtained from TPXO 9[37]. " - Is it commercially available or in-house?

9) In Page 3: "Siemens Wind Power A/S" - please be aware of that the company name is "Siemens Gamesa (SGRE). The name you are using is not any longer"

10) In Page 5: "?1 and ?2 represent the numbers of WA convector and TS turbine jointly installed with an OW turbine." - Regarding Table 1, please include an explanation of C1 less than 1. For example C1 =0.25 means one WA converter for 4 wind turbines. 

11) In Page 6: Between "Results" and "Output from single energy exploitations" - please add a short text such as The most energy efficient scenarios will be with reaching the highest mu at lowest V. This explanation will be good for better reading the results. 

12) In Page 7: ".We examined the outputs from the double-energy and triple-energy exploitations. " - Please remove the dot in front of this sentense.

13) In Page 7: "and recorded the corresponding indexes (Figure 6)" - Please explain what this "corresponding indexes" mean. Do you mean "relations" or "mutual effect"? A better explanation is needed here. 

14) In Page 7: "Principle 1 plus Principle 2" - please present this in a Table, for easier understanding of these two scenarios. 

15) In Page 9: "the high frequency OW power variabilities have large magnitudes and low predictability" - please include a typical period of such high frequency. Please give a time estimate of how fast can the wind power production change from zero to full power? In North Seas in Europe, it can change by less than an hour. It will be very useful to know how fast you consider in your study for your region. 

16) In Conclusion: "The investigation of this study reveals that in the adjacent waters of China, it is possible to increase the total amount of the integrated power output and decrease its temporal variations simultaneously by joint exploitations of offshore wind (OW), wave (WA), and tidal stream (TS) energy." - You could strengthen your conclusion, such as it is "possible and beneficial for the grid operation"

17) In Conclusion: "The comprehensive synergy of the triple-source energy exploitation is more significant than the OW-WA or OW-TS energy mining." - Consider this instead "The comprehensive synergy and less fluctuating power output of the triple-source energy exploatation  is more significant than OW-WA or OW-TS double-source generation." 

Reviewer 4 Report

The paper addresses an important issue, which is the co-located exploitation of three different sources of energy, so as to compensate for the daily variation that they have.

The study aims to characterise the different variability of the energy produced with the average value a coefficient of variation accounting for the variability of the resource. While this can be of interest, the most important aspects is to account for the correlation between the various sources of energy, which the paper mentions but does not exploit in full.

As wind energy produces much higher output  combinations of one wind turbine with other types of energy converters are considered. However wind turbines are deployed in farms (https//doi.org/10.1016/j.oceaneng.2020.107381)  to become economical and thus, solutions should be considered for such cases in which the number of the other type of converters may  become too many

The choice of the reference wave energy device is of importance. Wave Dragon was chosen for reference but it is not clear that this was the best choice, as this device has not become commercially available and has a rated power production much higher than other credible devices, which will distort the comparison. This can be shown in ( https://doi.org/10.3390/en6031344) where the power matrix of various devices can be found. At least other device should be considered to determine the influence on the outcome, as smaller output devices will require an increase in their number

The dayly variation of teh energigies is important and although the paper mentions it, it is not clear how this is modelled and how this is different for each of the nergy types. This should be discussed more extensively.

Round 2

Reviewer 4 Report

Reference 43 is incomplete